

# A machine learning, multi-band spectral reflectance clustering approach for examining physical transformations in landfast sea ice environments affected by spring freshets

Luka Catipovic[*,1,2,3], Samuel. R. Laney[1]
[1]Woods Hole Oceanographic Institution, Woods Hole, MA, USA
[2] The MIT-WHOI Joint Program in Oceanography/Applied Ocean Science and Engineering, Cambridge and Woods Hole, MA, USA
[3]City College of New York, New York, NY, USA

[*]Corresponding Author: Luka Catipovic, lcatipovic@ccny.cuny.edu

**Keywords:** coastal, Arctic, freshet, sea ice, machine learning, carbon

**Highlights**

- ML algorithms can classify water and sea ice characteristics in the coastal Arctic.
- Spring freshet runoff over sea ice is optically distinct from surface melt ponds.
- Time-series analyses of satellite imagery can reveal intra-annual freshet dynamics.





# Abstract

Spring freshets account for more than 50% of the annual terrestrial freshwater discharge into coastal margins in the Alaskan Arctic. Given the usual timing of Arctic freshets, much of this freshwater is discharged into coastal waters that are still covered by landfast sea ice formed the prior winter. This riverine freshwater both floods the sea ice surface and creates freshwater plumes immediately underneath landfast ice. We employed machine learning clustering algorithms to identify and characterize spatial and temporal variability in spring freshet overflows in the Alaskan Arctic, using the Sagavanirktok River as a model system. Multiband imagery from Landsat 8/9 OLI at the mouth of this river during the 2016 spring freshet were examined using the Caliniski-Harabasz method, which identified five unique clusters putatively representing areas of dry ice and snow, wet ice and/or snow, snow-free ice, ice-free open water, and areas of spring freshet overflow. A Gaussian Mixture Model algorithm, used to estimate cluster purity, indicated that the cluster representing freshet overflow is the most distinct from other clusters. Applying these approaches to an unusually comprehensive time-series of ten OLI images from 2022 revealed interesting spatial and temporal dynamics of these clusters as the freshet evolved, including the maximum spatial extent of freshet-flooded ice (271 km$^2$) occurring 2 weeks after peak estimated volumetric discharge, and the persistence of organic material-laden freshwater on top of landfast ice up to 10 km offshore until complete ice loss in early August.





## 1.0 Introduction

The presence of sea ice has important consequences for the physical environment of coastal

marine ecosystems in the Arctic. Sea ice blocks solar radiation in the fall, winter, and spring

(Light et al., 2008) and it presents a barrier to wind-driven mixing. It also provides organic

material to the underlying surface ocean during the melt season in late spring and early summer

(Smith et al., 1997), altering the abundance and species composition of pelagic bacterial and

seeding pelagic algal communities (Underwood et al., 2019; Olsen et al., 2017). Sea ice that

occurs within a few kilometers of shore is typically first-year ice that is anchored to the Arctic

coastline and shallow bottom regions. This ice is referred to as landfast ice and in the Alaskan

Arctic it typically forms over coastal deltas, lagoons, and estuaries completely by late October,

and persists until July or August, sometimes lingering into September (Weingartner & Okkonen,

2001).

One important annual event that affects landfast sea ice environments in many regions of the

coastal Alaskan Arctic is the spring freshet. Each year during late spring, the melting snowpack

on the adjacent terrestrial landmass is discharged into the coastal Alaskan Arctic via the

numerous rivers that drain the Alaskan North Slope, defined as the area between the Brooks

Range to the south and the Alaskan coastline to the north. These discharges represent large

volumes of relatively warm freshwater that flow onto and also immediately below the landfast

ice in coastal regions (Hearon, 2009; Okkonen & Laney, 2021). In rivers along the Alaskan

North Slope, the spring freshet typically occurs in May or early June of each year (Holmes et al.,

2008), and this event often accounts for more than 50% of the total annual freshwater discharged

into these coastal margins. Moreover, the freshet is arguably one of the most biogeochemically



significant annual events in these Arctic coastal margins, given the elevated concentrations of
dissolved and particulate organic material and nutrients contained in this terrestrially sourced
freshwater (McClelland et al., 2014). In addition to representing important contributions to
biogeochemical cycles in coastal waters, these amendments make the riverine freshet chemically
distinguishable from the ambient shelf waters found across the coast. Some of these chemical
differences can also be inferred from the optical signatures of these riverine waters, to indicate
qualitative and quantitative properties of said organic constituents (Catipovic et al., 2023).

The remoteness, scale, and optical characteristics of flooding and dispersal of freshet waters on
coastal sea ice make satellite remote sensing approaches potentially invaluable for examining the
spatial extent, temporal characteristics, and optical quality of any freshwater delivered onto
coastal sea ice during the spring freshet. Unfortunately, there exists at present no comprehensive
database of *in situ* matchup data of surface reflectance spectra of the freshet flood waters, which
creates significant challenges to developing remote sensing algorithms to study the seasonal
evolution of sea ice during the spring freshet. However, unsupervised clustering machine
learning (ML) algorithms provide valuable approaches for revealing spatial variability in satellite
remote sensing imagery for categorizing environmental phenomena. For example, ML methods
have found considerable value in classifying land cover (Paradis, 2022; Usman, 2013),
characterizing forest canopy constituents (Bunting et al., 2010), and identifying and mapping
agricultural crop types (Rivera et al., 2022). Specific methods, like Gaussian Mixture Models
(GMM) and k-means clustering, have been shown to be effective in partitioning features in
satellite imagery based on their reflectance properties in the visible and infrared portion of the
electromagnetic spectrum (Zhao et al., 2016). These algorithms likely have similar value for





classifying multivariate features such as the surface reflectance spectra of sea ice and coastal
Arctic waters. In this study we employed two ML clustering algorithms to optimize the surface
classification functionality of the United States Geological Survey (USGS) Landsat Operational
Land Imager (OLI). In particular, we focus on classifying and separating spring freshet overflow
from similar and surrounding ice and water types in a coastal Arctic lagoon system.

## 2.0 Methods

### 2.1 Study Area

This study examines the spring freshet of the Sagavanirktok River in 2016 and 2022, which is
delivered into Stefansson Sound in the Beaufort Sea along Alaska's Arctic margin. The
Sagavanirktok  is ~290 km long and drains a basin of nearly 15,000 km$^2$, with a watershed that
includes Arctic tundra as well as mountainous regions of the Brooks Range to the south that
result in a non-negligible watershed slope and a diverse network of tributaries (Connolly et al.,
2018). Spring freshets in the Sagavanirktok River are ephemeral, lasting on the order of one to
several weeks, but they can be particularly extreme, with a history of severely flooding the
Sagavanirktok River plain (Toniolo et al., 2017). Unlike the Kuparuk River immediately to the
west, whose freshet plume is channeled and contained by nearshore barrier islands, the freshet
waters of the Sagavanirktok River do not encounter barrier islands until ~ 15 km offshore and
thus spread predominantly northward as a single, large plume (Alkire & Trefry, 2006).

### 2.2 Satellite data acquisition and processing

The USGS Landsat satellite is polar-orbiting and its OLI utilizes a push-broom style multi-
spectral sensor configuration. This sensor includes five bands within the visible and infrared
range and collects 8-day repeat coverage over any given point on earth, including high-latitude



Arctic coastal regions. Level 2 collection 2 (C2L2) surface reflectance (SR) data from the OLI
on Landsat 8 and 9 were downloaded from USGS EarthExplorer. These data are atmospherically
corrected using the Land Surface Reflectance Code (LaSRC) and are subject to the known issues
over bright pixels which result in reflectance values outside theoretical limits (reflectance > 1.0)
(USGS, 2021). As such, these pixels were masked during processing. Data processing for the
analysis described here began with cleaning C2L2 scenes using the 16-bit quality assessment
mask provided within the downloaded data product. Pixels which failed the initial quality control
were flagged and not used in further analysis. Data were land-masked using the Global Self-
consistent, Hierarchical, High-resolution Geography (GSHHG) database of coastline polygons
(Wessel & Smith, 1996), with binary land masks created from the low resolution (51.2% of full
resolution) coastline data that were then applied to each scene to remove pixels representative of
reflection of light by land pixels. Additional outliers were removed from images using a z-score
method, where all surface reflectance values were converted to the standard normal form and any
pixels with an absolute z-score of 3 or greater were removed from the set.

We used a training image from May 12, 2016 at the mouth of Sagavanirktok River to explore
freshet plume overflow phenomena in detail (Figure 1). This Landsat 8 OLI image contains a
clear example of the freshet flooding on ice, and we exclude it from the 2022 time-series we used
for later ML analysis to ensure separation between training and testing data. This May image is
marred by a jet contrail shadow bisecting the lower third of the frame, and by a cloud shadow in
the northwest corner of the image. These features were not masked from the analysis due to
difficulties with masking clouds over ice. Their anticipated impact on the subsequent analysis is
discussed in section 4.1.






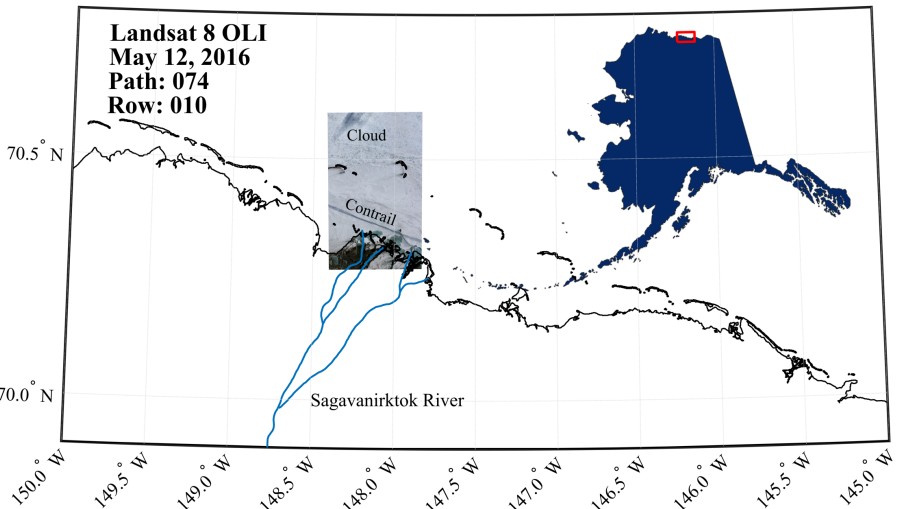

*Figure 1. Depiction of the Sagavanirktok River study area in context of the Alaskan Arctic coastline with superimposed Landsat 8 OLI Image showing freshet flooding over adjacent landfast ice on May 12, 2016.*

**2.3 Application of machine learning models to the Sagavanirktok River**
**freshet**
The Calinski-Harabasz (CH) (Caliński & Harabasz, 1974) method was applied to this imagery to
determine the optimal number of clusters for segmenting the data in any individual OLI scene.
The CH index provides a metric for determining how similar an object is to its own cluster (i.e.,
cohesion) compared to other clusters (i.e., cluster separation). Cohesion is determined by the sum
of the distances from data points within a cluster to that cluster's own centroid. Separation is
based on the sum distance from the centroid of the entire dataset, to all individual cluster
centroids. Surface reflectance bands one through five (B1 – B5) were tested for convergence, and
the set that resulted in the highest values within the allowed cluster range was chosen for further
clustering.




The k-means algorithm takes input in the form of the sample data and a specified number of
clusters, and then iteratively assigns cluster centroids until an aggregate minimum Euclidean
distance is reached between data points and centroids. This represents a simple clustering ML
model known as 'hard clustering' (MacQueen, 1967) that has been utilized in a number of
environmental applications (Sonnewald et al., 2019; Sun et al., 2021). For the current study we
used the KMean package from the Python Scikit-Learn toolbox (Pedregosa et al., 2011). One
important drawback of the k-means algorithm is that it provides no additional information
regarding the relationships between datapoints within separate clusters. To learn more about
these relationships, we employ a 'soft clustering' approach: the Gaussian Mixture Model
(GMM). These are probabilistic models that assume each data point is a member of a finite
number of Gaussian distributions, and the computational process is similar to k-means clustering
but provides additional information on the covariance structure of the data, allowing for a
comparison of likeness between clusters. For the analyses presented here we used the
GaussianMixture package from the Python Scikit-Learn toolbox (Pedregosa et al., 2011). Used
in tandem, these algorithms provide an avenue for cross-checking that each approach actually
converged to the global minimum within the solution space, while also providing complimentary
insight into the inner structure of the multi-band dataset.

Between April and August 2022, Landsat 8 and 9 observations over this region included an
unusual number of cloud-free scenes, which we leveraged to examine time-dependent changes in
the spatial extent of the clusters identified by these ML algorithms. Within this period, we found
ten cloud-free Landsat 8 and 9 OLI scenes from April 29 to August 25, 2022, where each scene



is level 2 collection 2 and is atmospherically corrected with the LaSRC and subject to the same
quality checks as the May 2016 training scene of the Sagavanirktok River mouth. During the k-
means analysis, pixels were assigned to the cluster they were closest to, as measured by
Euclidean distance to each cluster centroid. The spatial extent of each cluster was then computed
by multiplying the number of datapoints (pixels) within each cluster by the OLI sensor's 900 m$^2$
nominal area of each pixel.

## 2.4 Ancillary data

Time-series of air temperature and water temperature were obtained from observations collected
by the National Oceanic and Atmospheric Administration (NOAA Station PRDA2 -9497645,
70.41ºN, 148.53ºW), roughly 10 km to the west of the scenes used in this study. Volumetric
discharge of the Sagavanirktok River was measured by the USGS stream gauge 15908000
(69.01ºN, -148.82ºW), located roughly 80 km upstream from the mouth of the river. Volumetric
discharge values were extrapolated northward to the mouth of the river according to the scheme
described in Okkonen & Laney, 2021.

## 3.0 Results

### 3.1 Clustering OLI imagery during the Sagavanirktok spring freshet

Use of the Calinski-Harabasz method identified five optimal clusters within the May 2016
training scene when incorporating OLI bands ones through five, which are centered around 440
nm, 480 nm, 560 nm, 660 nm, and 850 nm respectively (Figure 2). Using additional bands
centered around longer wavelengths also resulted in convergence on five clusters, although the
CH scores at these peaks were lower than that of the peak associated with the first five bands.
Therefore, only the first five OLI bands were considered optimal for this clustering analysis.



Using fewer than five bands resulted in no convergence within the allowed criterion of 10
clusters.

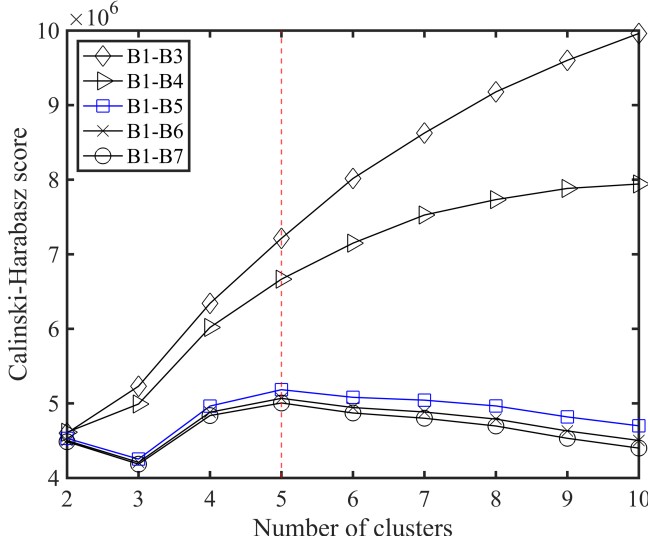

*Figure 2. Calinski-Harabasz evaluations for determining optimal number of clusters including Bands 1 through 3 up to Bands 1 through 7 from Landsat OLI. The dashed vertical line highlights Calinski-Harabasz scores using 5 clusters.*

The k-means and GMM algorithms assigned each pixel of the May 2016 training scene to one of
five clusters (C1 - C5) and provided the centroid spectra of that cluster. These centroid spectra
are the mean reflectance vectors of each cluster (Figure 3.), and each of the cluster centroids are
statistically different means from the other groups (Hotelling T-squared, $p < 0.05$ for all pairs).
In general, spectra associated with ice pixels (C1, C3, C4) are brighter than clusters associated
with water (C2, C5). Additionally, ice spectra reflect most in the blue, whereas water spectra
tend to reflect most in the green portion of the spectrum. Cluster centroids match well with
reflectance spectra of various ice and water types that have been directly measured in the central
Arctic Ocean (Istomina et al., 2016) and are documented in the PANGEA database, as well as
with spectra measured near the mouth of the Mackenzie River (Klein et al., 2021). C1 was most





similar to fine white ice (PANGEA ID: 260812purice1e00000), while C3 resembled ridged ice
(PANGEA ID: 110812ROVtransect17e00000). C4 closely resembled a dark melt pond with a
thin layer of ice on the top (PANGEA ID: 150812ROV67pbwe00000), however the melt pond
spectrum was slightly depressed in the blue region suggesting that the spectral averaging to
achieve the C4 centroid included more ice pixels than ponded pixels. C2 closely resembled the
reflectance of open water measured 50 m offshore near Herschel Island Qikiqtaruk at the mouth
of the Mackenzie River in Summer 2019 (Klein et al., 2021), but is comparatively elevated in the
red region of the spectrum. We were unable to find measured spectra of freshet overflow over
landfast ice, and so have no comparative measure for the C5 centroid. However, based on first
principles of reflectance of case 2 waters with high concentrations of organic material and the
reflectance of melt ponds, C5 putatively represents the region expected to be occupied by freshet
flooding over ice.

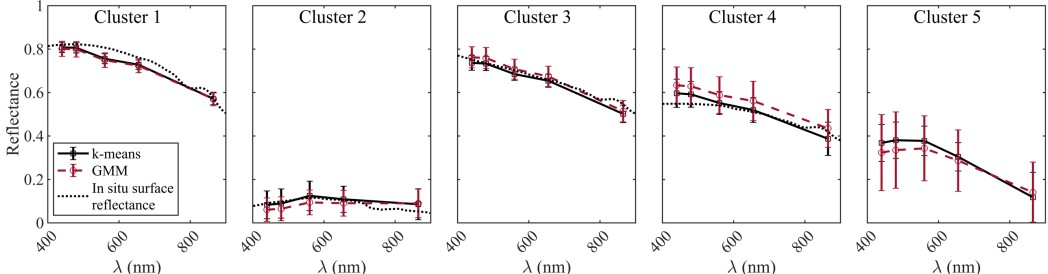

*Figure 3. Cluster centroids as determined by k-means analysis (black squares) and a Gaussian mixture model (red circles). In situ reflectance (dotted lines) are provided for comparison, see text in section 3.1.*

When plotted on geographic axes, the clusters are generally grouped spatially, with some
concentrated in a single region while others more spread out (Figure 4). C1 and C3 occupy areas
in the northern portion of the scene that are expected to contain only sea ice. C2 occupies the
area expected to contain open water at the mouth of the river to the south, C4 occupies areas



expected to contain snow-free ice and/or melt ponds as well as any clouds, and C5 occupies the
area expected to contain freshet overflow at the interface between open water and ice.

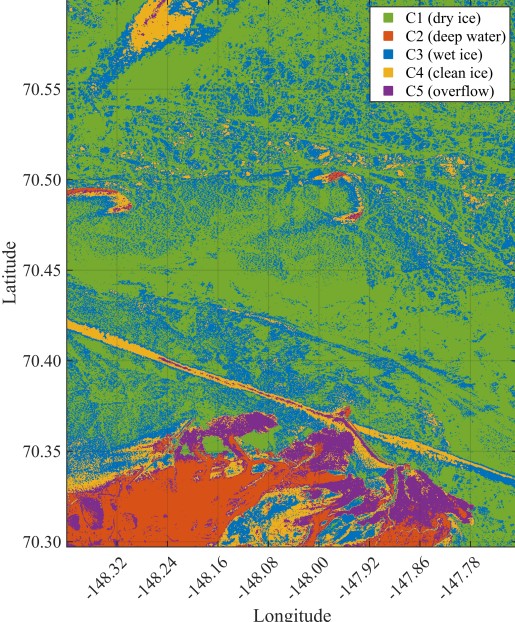

*Figure 4. False color image of the 5 clusters identified by k-means in the Landsat-8 OLI training scene (May 12, 2016 (Path:10 Row:74). Landsat-8 image courtesy of U.S. Geological Survey.*


## 3.2 Statistical assessment of cluster uniqueness

Similar to k-means clustering, the Gaussian Mixture Models algorithm partitions multivariate
datapoints into clusters based on their Euclidean distance to each cluster's centroid. However,
each datapoint may lie within the normal distribution of one or more other centroids. This creates
a probability distribution of clusters to which that specific point may be assigned. In some cases,
the probability of a point being associated with a given cluster is just barely over 50%, which
indicates mixed belonging, but in other cases it is 100%, indicating absolute belonging (Figure
5). Cluster 5 contains the most data points that strictly belong to that particular cluster, but the



standard deviations around each reflectance value in the centroid of C5 are larger than for any
other centroid (mean std. dev. of 0.16 across all bands compared to 0.03, 0.06, 0.05, and 0.09 for
clusters 1 through 4 respectively). Thus, the distribution encompassing the C5 data points is the
widest (i.e., encompassing the most variability within assigned pixels), but also the most unique
from other clusters.

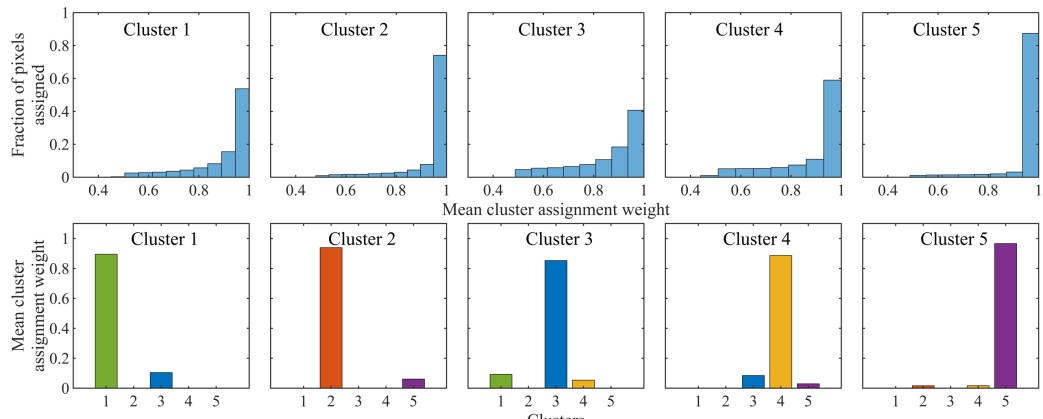

*Figure 5. Top row: Distribution of weights used to assign pixels to each cluster by a Gaussian Mixture Model. Bottom row: mean cluster assignment weight of each pixel assigned to a given cluster.*

Each cluster is populated by a significant number of data points having an assignment weight
above 75%. Moreover, data points under this requirement account for over 85% of all data points
in the set demonstrating good separation of clusters within the Gaussian Mixture Model. The
mean component weight for a pixel to be assigned to a given cluster is 0.89, 0.94, 0.85, 0.89, and
0.97 for clusters one through five, respectively. Cluster 5 accounts for 100% of all data points
with 100% belonging and 80% of all pixels in C5 were assigned by a value of 0.99 or more
(Figure 6). The two clusters with the brightest centroid spectra (C1 and C3) coincide with areas
in these Landsat OLI scenes where ice and snow are expected. These clusters share the greatest
proportion of pixels within each other's distributions as identified by GMM. In each assignment





for C1, a given pixel is 10% similar to C3 on average, and each pixel in C3 is on average 9%
similar to C1. Overall, these two clusters contain the greatest number of pixels with the lowest
weights.

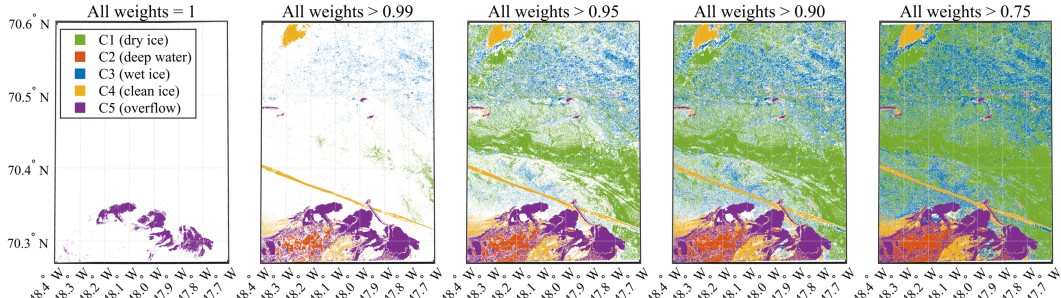

*Figure 6. Successive inclusion of descending weight values used for assignment in each cluster calculated by Gaussian Mixture Model. Landsat-8 image courtesy of U.S. Geological Survey.*

## 3.3 Temporal dynamics of the spring freshet and landfast ice in Stefansson Sound

By applying this ML approach to a sequence of OLI images, a time-series analysis can be
performed that reveals temporal changes in the spatial dynamics of clusters assigned to ice and
water types at the mouth of the Sagavanirktok River during the 2022 spring freshet event and the
following summer months (Figure 7). The series starts in late April with a scene completely
covered by cluster 1, which is representative of dry ice, and it ends in late August with a scene
completely covered by cluster 2, which is representative of open water. The dynamics of the
other clusters within the intervening period provide insight into the timing of when the
intermediate clusters appear and the extent to which these clusters occupy areas within the given
region. As the dry and wet/ridged ice clusters melt away, the overflow cluster (C5) and the snow-
free/ponded ice cluster (C4) increase in area in unison before open water occupies the majority
of the image. The cluster representative of spring freshet flooding over ice (C5) reaches a



maximum areal extent on June 16, 2022 with a total coverage of 271 km$^2$ within the full Landsat
OLI scene.

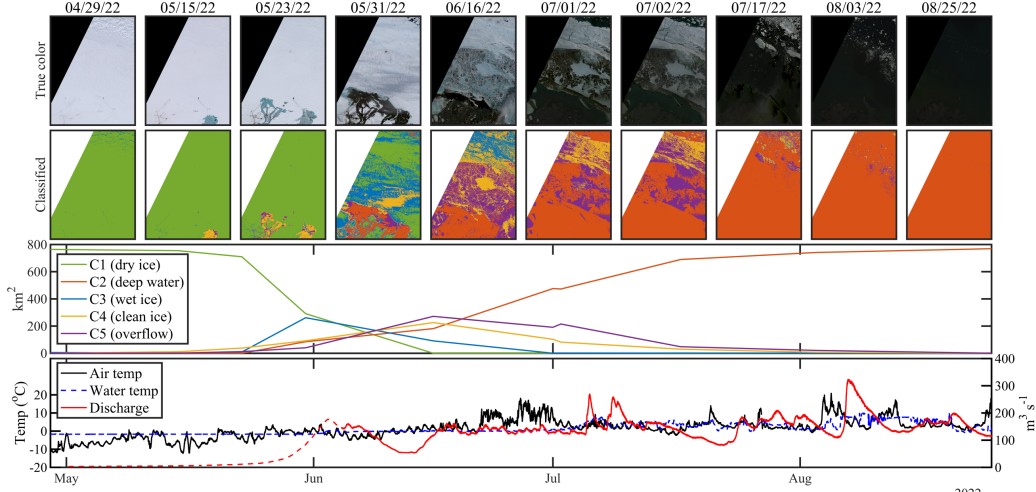

*Figure 7. Time-series of the mouth of the Sagavanirktok River and nearby areas of Stefansson Sound from April 29, 2022 to August 25, 2022. This time span encompasses the spring freshet as well as the subsequent summer ice melt dynamics. True color RGB images (top panel) show progression from ice covered coastal margin to open water. ML classified false color images (second panel) show progression from complete coverage by cluster 1 (ice) to coverage by cluster 2 (open water). The line plot in the third panel shows evolution of area occupied by each cluster as identified by k-means analysis throughout this period. The bottom panel shows air temperature, ocean water temperature, and volumetric river discharge from data sources mentioned in section 2.4. Dashed discharge in late spring indicates estimated discharge. Landsat-8/9 images courtesy of U.S. Geological Survey.*

Understanding the dynamics presented in this time-series involves an awareness of the relative
spatial changes in air temperature between clusters over the course of the time period. As
ambient atmospheric temperatures begin to warm in the late spring, the dry ice cluster (C1)
begins to become less ubiquitous in favor of a cluster resemblant of wetter ice (C3).
Concurrently, snow-free/ponded ice (C4), open water (C2), and freshet overflow (C5) begin to
occupy the mouth of the river following the onset of the spring freshet, as indicated by the rapid
increase in riverine volumetric discharge. As melting and flooding continue, C5 and C2



increasingly encompass the scene until late August, when all ice and remaining flooding is no
longer visible. From the first appearance of open water at the end of May 2022, when it covered
just 84.8 km$^2$ of the 973 km$^2$ region, three months elapse before open water encompasses the
entire region contained in this OLI image series.

## 4.0 Discussion

This study highlights several important advantages to applying unsupervised ML clustering
algorithms to satellite remote sensing data products in Arctic coastal margins. Using a ML
approach improves our understanding of the distribution of ice and water features throughout the
melt season and provides an avenue for identifying the timing and extent of on-ice flooding
during the spring freshet. This study demonstrates how the Calinski-Harabasz method can be
used with multi-band remote sensing surface reflectance data to identify an optimal number of
distinct clusters within a multivariate image. This approach not only identifies the ideal number
of clusters to input into an unsupervised clustering algorithm, but it also determines the best
subset of data to use in order to create disparate clusters. In this case study, the Calinski-
Harabasz method determined that using the information contained in additional bands resulted in
less accurate separation between identified clusters. This represents valuable enhancement in
data preprocessing and exclusion when investigating statistical relationships between reflectance
spectra within remote sensing data sets.

Unsupervised machine learning clustering algorithms were also shown here to successfully
partition surface reflectance data of an Arctic coastal margin into the five unique groups as
identified by the Calinski-Harabasz method. These groups are represented by centroid spectra
that show significant similarity to *in situ* surface reflectance spectra that have been measured





directly from various Arctic regions by other workers. Although the measured spectra are not
local to the part of the Arctic examined in this study, these measurements nonetheless represent a
useful form of ground truthing that promotes further analysis regarding the creation and fate of
each cluster within our time series. Finally, to the best of our knowledge, this specific time series
represents the first remote sensing analysis of the entire seasonal trajectory of a spring freshet
flooding event at an Arctic coastal margin.

## 282    4.1 Advantages of using k-means and Gaussian Mixture Models to examine

## 283    multi-spectral remote sensing data products

The k-means and GMM algorithms offer distinct advantages for using remote sensing imagery
from sensors such as OLI to interpret basic seasonal transformations in coastal Arctic regions.
Both of these ML algorithms are unsupervised, meaning there is no predefined output variable to
be mapped onto, in contrast to supervised algorithms which simply create a mapping function
from a set of input data to a set of output variables. Another key benefit of employing
unsupervised algorithms to large data sets is the insight these algorithms provide about the
underlying structure or distribution of the input variables, which can be grouped according to
their mathematical likeness. This methodology becomes highly effective when paired with
remote sensing imagery, as multi-band sensors generate a hypercube of variables that can often
be relatively independent. Additionally, often there exists no accompanying set of *in situ* data
with which to train a supervised model, which leaves unsupervised methods as the only feasible
option for interpreting spatial and spectral patterns in remotely sensed imagery such as the OLI
observations we examined here.





From an environmental perspective, in a system such as an ice-covered Arctic coastal margin it
can be valuable to identify which clusters or groups exist within the system and to define them in
terms of their derived spectral signature, rather than assigning pixels to predefined features.
Gaussian Mixture Models excel at providing this sort of insight into the relationships between
derived clusters. Its soft clustering approach assigns a Gaussian distribution around each cluster
centroid, which is inclusive of pixels not assigned to that cluster. This method retains
information regarding how similar clusters are to one another, where for example clusters with a
high proportion of pixels also existing within the first or second standard deviations of another
clusters' distribution are more similar compared to those that share no pixels. This information is
requisite for inferring actual environmental relationships between similar entities, such as those
between dry ice and wet ice or ponded ice and snow-free ice.

One final advantage of utilizing unsupervised clustering algorithms with remote sensing imagery
is the retention of the multivariate spectral units in the output. This helps to strengthen the
relation of identified clusters to measurable features such as surface reflectance, and thus allows
for the cluster centroids to be interpreted as representations of real features. Such interpretations
are key to examining spatial and temporal dynamics in remote regions of the world such as the
Arctic and linking them to relevant environmental phenomena.
**4.2 Application of k-means and Gaussian Mixture Models to river-fed sea ice**
**systems in the coastal Arctic**
Arctic coastal margins are ideal environmental systems for exploring how unsupervised
clustering algorithms can be utilized with remote sensing data products. Extreme seasonality



renders these regions difficult to sample directly to collect the *in situ* data required for training
supervised ML algorithms. Moreover, these unsupervised models offer a novel way to tag and
track unique features of water and sea ice that are exclusive to coastal Arctic regions receiving
riverine input. In this study, k-means and GMM models were able to identify a cluster
represented by a spectrum that bears resemblance to the expected spectra of freshet water
overlaying sea ice, a feature previously unstudied with respect to *in situ* reflectance
measurements. By identifying important features such as freshet flooding on ice, ML tools can
help answer questions regarding how long freshwater persists on sea ice, and how far seaward
this on-ice flooding reaches. Such information would be highly valuable, for example, in studies
that examine how on-ice flooding melts overlying snow cover and thus allows more light into the
water below, which has important biogeochemical consequences to under-ice photosynthesis and
photodegradation of riverine organic material.

The identification of these clusters represents a first-of-a-kind approach to assessing the temporal
evolution of Arctic coastal ice features throughout a highly dynamic spring-summer melt event.
One unique outcome from applying these ML methods in this study was the time-series analysis
that used OLI imagery to track the appearance and disappearance of individual features
associated with seasonal changes in these coastal Alaskan Arctic waters. From April 29 to May
23 in 2022, areal coverage of Stefansson Sound was primarily represented by cluster C1,
putatively dry ice or snow-covered ice due to its bright reflectance spectra. On May 15, there was
a slight intrusion of C4 (snow-free ice) in the southeast corner of the scene at the mouth of the
Sagavanirktok River, suggesting the emergence of liquid water flowing from the river's mouth,
which resulted from upstream snow melt due to warmer temperatures in the southern reaches of



the North Slope. In this May 15 scene, the appearance of C4 reflected a phenomenon where
water flooded over ice, melted surface snow, and smoothed the ice surface before it drained back
through strudel holes. The appearance of this smooth, snow-free ice represented the first phase of
the spring freshet in Stefansson Sound. Concurrently, as the spatial coverage of dry ice in
Stefansson Sound declined due to melting, the region covered by wet/ridged ice increased in area
slightly before, in turn, beginning to decline. This all happened when average air temperatures
started to remain above freezing. From this we can infer that C3 represented the expected wet ice
that reflected light in a different manner from the snow-covered ice (Vérin et al., 2022).
Furthermore, the decline of these two clusters coincided with a substantial warming event that
triggered the full onset of the freshet as indicated by peak volumetric discharge. In late May, C2
(open water) began to appear at the mouth of the river while C5 (freshet overflow) continued to
distribute over lingering sea ice. Peak coverage by C5 occurred on June 16 which also
represented the last scene containing C3. The 6-week period from May 31 to July 17 showed that
river flooding on ice occurred well into the summer and had substantial implications for sea ice
melt.

**4.3 Caveats and concerns of using clustering models on remote sensing data in**
**such applications**
While clustering algorithms offer a promising avenue for studying Arctic coastal margins, there
are some noteworthy caveats and concerns that warrant consideration. *A priori* knowledge from
other parts of the Arctic allows us to infer the meaning of clusters based on the similarity
between their centroids and *in situ* surface reflectance data observed directly, yet these



interpretations remain purely an inference. Moreover, we recognize that while the Calinski-
Harabasz method mathematically predetermines an ideal number of clusters for the specific input
data set, there are a large number of subdivisions of such data that could be made for any number
of expected features, which can introduce artifacts. An example of this might be seen in a
situation where three clusters were identified as unique ice types, when in fact one was
mislabeled and instead represents standing water over ice. We recognize these limitations and
encourage future users to not apply such unsupervised ML models with the expectation of
identifying a set of features, but instead to use the features found by the algorithm as a starting
point for examining the system under study.

We are also cognizant of potential problems that can arise when evaluating the purity of clusters
that are identified via unsupervised ML models. The GMM output provides information on the
interrelationships between clusters, but it provides no information on any intrarelationships
within clusters. This issue arises most notably when pixels are mathematically similar to each
other in terms of their Euclidean distance, but are actually distant in an environmental context.
One example of this in our data set occurs with the spectral similarities between clouds and
certain types of sea ice. Cloud masking of visible imagery that contain sea ice is a longstanding
challenge for Arctic remote sensing (Istomina et al., 2020), and the ML models used in this study
provide no advantage to discriminating clouds from terrestrial features. Specifically, our May
2016 training image contains two atmospheric features that were misidentified as representing
our cluster C4: a cloud in the northwest corner of the image and a jet contrail horizontally
bisecting the southernmost third of the image. While these represent features that can be
identified readily through visual inspection, clustering algorithms are unable to separate them



into their own group. We suggest revisiting this limitation as hyperspectral data products become
more widely available, as their higher spectral resolution enables better differentiation between
features with similar spectral properties.

While the Landsat OLI sensor offers a high-resolution footprint of 30 m x 30 m, spatial
heterogeneity near the edge of features and within features nonetheless still affects the average
spectra representing each cluster. When a single pixel includes multiple features, the resulting
spectral darkening or lightening will ultimately affect the assignment of pixels to a cluster. This
phenomenon is particularly problematic near the edges of large, spatially dense clusters, or
within speckled groups of similar clusters (Dantas de Paula et al., 2016). In the coastal Arctic
regions we examined, specific examples of this phenomenon involve the brightening of the red
reflectance in C2 (open water) near the mouth of the Sagavanirktok River, which we believe is
likely an artifact of the inclusion of bare land in the nominally open water pixels. Additionally,
the braiding of rivers that occurs within the Sagavanirktok River, as well as other rivers along the
Alaskan Arctic coast, exposes river bed between streams of running water. This phenomenon
occurs on spatial scales smaller than the OLI pixel footprint, and therefore pixels that might
actually be freshet overflow may be grouped with C4 (snow-free / ponded ice) if they encompass
water that is too shallow, or with C2 (open water) if the water is too deep.
**4.4 Blue-band ratios as an indicator of CDOM absorption in cluster spectra**
A central outcome of this study is the ability to discriminate the type of water that overlies
landfast sea ice: spring freshet waters introduced by flooding versus melt ponds created by local
melting. These waters are biogeochemically different, with freshet waters containing
considerably higher amounts of organic matter and suspended particulate material compared to



coastal oceanic waters (Holmes et al., 2008; McClelland et al., 2014). Organic matter absorbs
light exponentially with decreasing wavelength (Kirk, 2010) and particulates tend to block solar
radiation, which together will lead to waters with higher concentrations of organic material
having lower ratios between the reflectance values within OLI's two blue bands (B1/B2). These
blue bands are relatively close together spectrally and thus are likely to be impacted similarly by
water depth. This limits the possibility for using these two bands to identify any differences due
to the presence of organic material. This absorption by organic material in freshet waters is
therefore likely manifested in the reduced reflectance in the blue wavelengths in the C2 cluster,
which has been labelled open water, and the C5 cluster, which has been labelled freshet water
over ice.

Additional insight into the utility of this blue-band ratio for examining potential differences in
CDOM content of waters in this region can be gained by inspecting transects that cross clusters
in these images (Figure 8). A transect across an area of C4 pixels (blue ice/melt pond, Figure 8A)
is framed by inclusions of C1 and C3 (dry and ridged/wet ice, respectively), and within the blue
ice/melt pond region, the surface reflectance values decrease compared to those within the dry
and ridged ice regions while the blue band ratio remains roughly at unity. However, across the
transect from C2, to C5, to C4, to C1 and C3, near the mouth of the river (Figure 8B), there is a
distinct increase in blue-band ratio, suggesting the influence of the spectral absorption signature
of organic material on the reflectance of these leading pixels. While this is an early indication of
the possibility for detecting organic material in flood water on ice, this type of analysis offers a
promising path forward for investigating such phenomenon in other Arctic regions. Moreover,
while other remote sensing tools such as synthetic aperture radar can detect water on ice given



that the surface is perturbed sufficiently to scatter radar waves (Hearon, 2009), such observations
offer no insight into the biogeochemical quality of the water, which would differentiate between
melt ponds and freshet flood water.

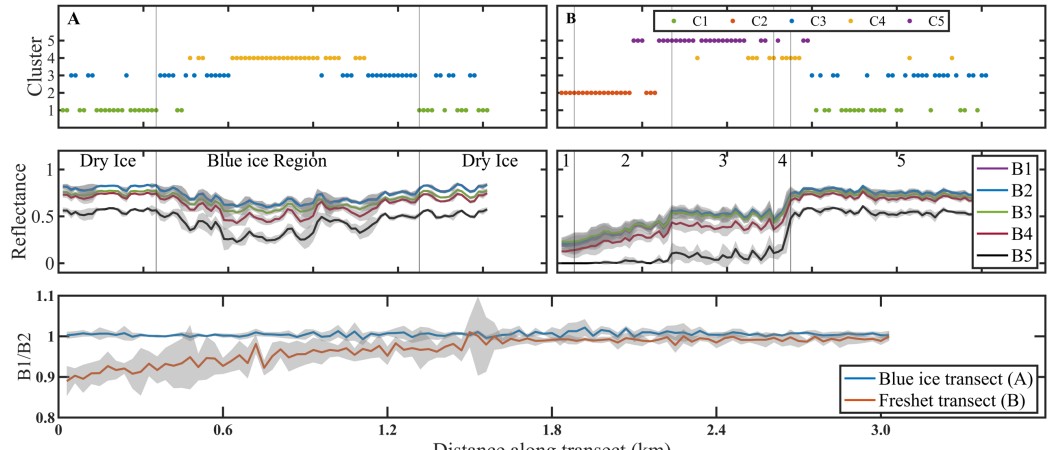

*Figure 8. Top row: a 3 km transect across A) a melt pond region (C4; yellow) and B) a freshet flood overflow region (C5; purple). Middle row: multi-band reflectance of distinct regions within each transect: 1) open water, 2) heavy flooding over ice, 3) stable flooding over ice, 4) transition period from flood to dry ice, 5) dry ice. Bottom row: Band 1/ Band 2 ratio of each above transect.*

## 4.5 Future directions

Findings from this study open several avenues for future research that could utilize unsupervised
clustering algorithms with remote sensing data products to investigate sea ice and flood water in
Arctic coastal margins. First, the approach outlined in this study is not limited to any individual
sensor on any one platform. Sentinel-2 Multi-Spectral instrument (MSI) also measures a
sufficient number of wavebands to identify multispectral snow, water, and ice features based on
their spectral identities (Buckley et al., 2023). This sensor has a similar yet offset revisit period
to that of OLI, allowing for increased temporal resolution while maintaining a high level of
spatial detail. However, unlike fractional snow/ice cover, which can be interpolated in time based



on meteorological conditions (Zakeri & Mariethoz, 2024), the spring freshet is a highly
ephemeral event that has much more complex daily variability. Therefore, there is also merit to
translating these unsupervised clustering algorithms to sensors with higher temporal resolution
such as Sentinel-3 Ocean and Land Colour instrument (OLCI). OLCI has been used in a similar
context to this study for identifying and tracking land ice properties based on spectral reflectance
(Kokhanovsky et al., 2023), and could supplement this research by providing more accurate
estimates as to the time freshet water spends on ice before presumably draining through strudel
holes to the ocean below.

Additionally, unsupervised clustering methodologies can be applied to images from sensors with
much finer spatial and spectral resolution, such as NASA AVIRIS-NG and USGS Hyperion, to
discriminate between spectrally similar ice types (Han et al., 2017). Spectrally, clustering of
hyperspectral resolution data products results in cluster centroids with more defined spectral
shapes, allowing for more accurate definition of centroid identities. One drawback of our
approach here is the inability to discriminate between ice features that have similar reflectance
spectra, given the bands provided by Landsat OLI. The finer spatial and spectral resolution from
AVIRIS-NG could address this gap by providing the ability to account for differences in similar
ice types through changes in reflectance spectra due to varying snow grain size and snow layer
content (Nolin & Dozier, 1993; Rosenburg et al., 2023). This further level of classification would
decrease the ambiguity between wet and ridged ice, sparse and dirty ice, and clean ice and melt
ponds. More generally, a multi-sensor approach would greatly strengthen the utility of the
methodologies outlined in this study by extending beyond the spatial, temporal, and spectral
limitations presented by any single sensor system.






Environmentally, the approaches in this study are directly applicable to biogeochemical studies
regarding the delivery of organic material and heat energy to coastal Arctic Ocean ecosystems.
The spring freshet is a major source of organic material for the coastal Arctic Ocean, and its fate
is of core interest in Arctic marine biogeochemistry (Stedmon et al., 2011). This organic material
spreads across the surface of the sea ice and is exposed to high levels of solar energy which alters
its chemical composition via photo-oxidation (Cory et al., 2014; Grunert et al., 2021). These
compositional shifts would imply a reduction in the amount of labile organic material exported to
the surface ocean from previously estimated values from riverine discharge (Holmes et., 2008),
which may increase primary productivity estimates with respect to heterotrophic bacteria and
phytoplankton competing for nutrients (Thingstad et al., 2008). Unsupervised clustering
approaches could provide an avenue for determining the residence time of water on ice before
draining through strudel holes, thus offering insight into the extent of photo-degradation
occurring before said organic material reaches biological communities in the water column
below. Additionally, the spring freshet transports a large amount of heat to the surface of coastal
sea ice (Okkonen & Laney, 2021), which melts snow resting on the surface and in turn
drastically changes the transmissivity of the snow-ice column (Redmond Roche & King, 2024).
While the increased light transmitted is unlikely to further photo-oxidation of the freshet waters
flowing under the ice, it may allow for increased rates of ice algae production (Hill et al., 2022).
The area of cleared ice, identified by C4 in this study, may promote remote sensing evaluations
of sub-ice biomass via estimations of transmitted irradiance through snow-free ice (Ardyna et al.,
2020), or bare ice surrounded by melt ponds (Laney et al., 2017). This knowledge would offer



valuable new insight into the biogeochemical implications of the spring freshet as it relates to
Arctic coastal ecosystems.

Other interesting potential applications of unsupervised clustering of remote sensing reflectance
in Arctic regions may lie in areas outside of oceanography and geoscience. Ships rely on
knowledge of ice conditions to safely traverse across polar ocean waters, and icebreakers in
particular can leverage snow cover data for safe wayfinding given that snow cover has been
shown to increase both breaking and submersion resistance for icebreakers, making it an
important variable to map for successful travel (Huang et al., 2018). Machine learning clustering
algorithms can support these societal sectors by providing maps of ice surface conditions across
the entire Arctic. This information is also critical for facilitating Arctic shipping and trade as
often icebreaking escorts are required to cross Arctic environments. In addition to assisting
Arctic safety and shipping operations, optical remote sensing is also a key technology identified
for efficiently detecting and responding to oil spills on ice (Palandro & Mullin, 2017), which can
significantly impact subsistence hunting and animal habitat (Wilson et al., 2024). Similar
approaches can be used to monitor oil spills in the open ocean (Haule et al., 2021), which can
have similarly drastic sociocultural, economic, and ecological consequences (Gill et al., 2016;
Barron et al., 2020). If applied carefully, unsupervised clustering of these events could greatly
improve response time and environmental impact of such environmental catastrophes. An
additional socioeconomic application for clustering of optical remote sensing data is the
identification of harmful algal blooms. Many harmful algae species have unique optical
characteristics that make them differentiable from each other and from surrounding water types
(Gernez et al., 2023), especially within heterogenous distributions across coastal margins



(Caballero et al., 2020). Knowledge of species, location, and extent, could be used to inform
policies set for public safety such as fishing regulations, beach closures, and shellfish
consumption.

**CRediT authorship contribution statement**
**LC:** Conceptualization, Investigation, Data curation, Methodology, Software, Formal analysis,
Visualization, Writing – original draft, Writing – review & editing. **SRL:** Funding acquisition,
Conceptualization, Investigation, Methodology, Formal analysis, Writing – review & editing,
Resources, Supervision

**Competing Interests**
The authors declare that they have no conflict of interest.
**Acknowledgements**
This research was funded by the National Aeronautics and Space Administration's
Interdisciplinary Research in Earth Science (IDS) program (award NNH19ZDA001N-IDS).










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
