# Peer review of "A machine learning, multi-band spectral reflectance clustering approach for examining physical transformations in landfast sea ice environments affected by spring freshets"

_EGUsphere, 2025_

## Referee Comment (RC2)

Review of
**A machine learning, multi-band spectral reflectance clustering approach for examining physical transformations in landfast sea ice environments affected by spring freshets**
https://doi.org/10.5194/egusphere-2025-1450

The manuscript describes an approach to using unsupervised ML clustering algorithms to measure extent of the spring freshet where a river meets Arctic sea ice. The argument that unsupervised ML algorithms have substantial potential in this area is somewhat undercut by the lack of validation of the clustering/classification and some dubious classifications in both the training and the time series imagery. While the concept is compelling, the implementation has room for improvement.

**Major comments:**

**Inaccurate surface classification:**
The approach of matching spectra of the clusters to surface types (discussed in 182-190) is either inappropriately simplistic or insufficiently explained. As these clusters cover a large area with varying surface types (based on a visual inspection of the image), picking an individual spectral profile as a "match" for each cluster seems unlikely to be accurate. Was this done manually? I expect for most if not all clusters, more than one measured spectral profile fell within (or mostly within) the spectral bounds. Including the full range of surface types that match a cluster (ideally in addition to measurements at more locations/times of year) could help with some of the issues around inaccurate surface classification. If you already did this, make it clear in the text.

Figure 7 shows several dubious classifications, especially suspicious when you compare the true-color imagery to the classifications:
• Based on the visible imagery and the shape of the features, C4 (clean ice) seems like a better classification fit for fresh water overflow than C5.
• In the 5/31 image, an area classified as overflow (slightly triangular shape near lower center) looks more like thin cloud over open water or snow-free land. Shadow patterns in the image would confirm the presence of thin clouds.
• Starting in the 7/1 image, there are areas (e.g., center of the image) that are classified as open water but on close inspection have cracks and edges and other features associated with ice rather than open ocean.
• There are features in the 7/1 and 7/2 image (e.g., center of the lower half of the of the image, along the straight ice edge) that are classified as overflow, but the spatial pattern does not make sense for ponded overflow lingering on the sea ice for weeks. Especially the blob in the center, which is identified as clean ice in 6/16 and then without substantially changing shape, classified as overflow in 7/1-2.
• In the 7/17 and 8/3 images, what seems like rotten ice is classified as clean ice or overflow.
• Land is classified as dry ice, clean ice, and deep water.

**Training image:**
Was the training image sufficiently late in the season to include the different surface types relevant to the later images? For example, an early May training image would have included snow-covered or at least mostly frozen land surface, while later summer images had a fully thawed land surface. This might have contributed to some of the issues with inaccurate surface classification.

Cloud masking is clearly a problem: why is known cloud/jet trail included in the "clean ice" classification? It seems like removing these areas from the training image might improve the classifications overall.

Likewise, land is not masked in the training image. Much of what is classified as "C2: deep water" and "C5: overflow" in the training image is clearly area that should be excluded based on the land mask described earlier in the paper.

How is cloud filtering handled in this and other images? Does anything account for sun angle?

**Validation:**
There is no description of any validation of the surface type assignments to the different clusters in the paper: given the issues with inaccurate surface classification, at a minimum there should be some visual inspection to make sure the classification names match what is visible in the true color images.

Without validation of the classification, many of the statements in the discussion are somewhat questionable. Specifically, figure 7 indicates that overflow from the freshet externs through the summer into early August. The detailed narrative of the season starting at line 333 depends on accretive classifications, but without validation of the classifications that is a big unknown.

**Land masking:**
Section 2.2 describes an approach for land masking, and figure 1 shows the training image with a coastline overlay, but the land masking does not appear to be applied in any further figures.

**Paper organization:**
For a methods development paper, the distinction between "Methods" and "Results" can be a fuzzy one. Renaming and breaking up the sections (Methods -> Data; Results -> Classification algorithm, Results ) would make it easier to follow. Likewise, splitting the Discussion section into Discussion and Conclusions would help.

The detailed narrative of the seasonal progression (starting several sentences in to the paragraph at 333) should be provided closer to the time-series figure (7) as it's own section.

**Figure comments:**

| Figure numbers | Comment |
| --- | --- |
| Figure 1 | It is hard to tell with this small an image, but it seems the coastline in the image and the coastline outline do not actually line up (for example, the islands do not line up). This suggests problems with the geolocation approach, and therefore the landmasking. The scale of map is unnecessary for this paper, but getting a better look at the true-color image for comparison with figure 4 is important. I recommend getting rid of figure 1 (see comments on figure 4). If you are going to keep figure 1, add the river to the inset Alaska map to provide a sense of the scale of freshwater runoff catchment. |

| Figure 2 | Given the normal readership of The Cryosphere, additional explanation of the CH score and how to interpret figure 2 is necessary. Based on context I infer that the optimal is the number of clusters and bands where the CH score starts to drop, but some actual explanation would be helpful. |
| --- | --- |
| Figure 3 | Clusters 1, 3, and 4 look very similar in this presentation. Maybe adding shaded regions for the other clusters to the background of each spectral signature would help distinguish them? You also color-code the clusters in later images. Maintaining that color-coding throughout the figures would help.
As in the comment on inaccurate surface classification, adding more in situ measurement lines (and labeling them) would make the classifications seem more robust. |
| Figure 4 | Adding the true color image side-by-side for comparison would be helpful in understanding how well the classification works. |
| Figure 5 | Do the statistics presented in this figure hold for a different image from the original training image? Are clusters from another image later in the season similarly distinct? |
| Figure 6 | This image sequence does not have the and mask applied – and it seems almost all of the highest weighted assignments are either on land or are the cloud/jet trail features. |
| Figure 7 | The land mask does not seem to be applied to these images - this is causing an area to be mis-classified as open water when it seems (based on visual inspection of the image) to be unfrozen land surface for much of the year.
The third panel would benefit from markers to indicate which dates have data and which are interpolation.
The axis range on the temperature/discharge plot is so limited, it is hard to read. Shading the "melt season" when air temperatures are >0 might help legibility. |
| Figure 8 | Adding a slice of each image as a top panel would help see what is being labeled as each of the clusters, as well as make sense of the reflectance data in context.
The top two rows of panels need x-axis labels, and if the bottom panel is meant to correspond to each of those, it should be split in two and shown on the same x-axes rather than combined.
What do the numbers indicate on the B-reflectance panel? |

**Minor comments:**

| Line number(s) | Comment |
| --- | --- |
| 32-36 | The timing of ice melt/freezing on Arctic coastlines provided by this reference is >20 years out of date. |
| 46 | Provide a reference for the "more than 50%" estimate. |
| 122-125 | Discussion of the CH index is hard to follow – given the spatial aspects of this analysis, be clear when distance is referring to spectral distance (or distance in spectral space?) rather than physical distance. |
| 160 | What is the purpose of the water temperature data from a location that would not be impacted by the freshet? |

| | |
|---|---|
| 165 | A brief summary of the river discharge extrapolation approach would be helpful here. |
| 193-197 | There seem to be a lot of assumptions made here - based on what first principles? Why does 5 seem to be the freshet flooding? Be explicit in your reasoning. |
| 230 | The discussion of statistical separation of clusters would benefit from including a second image in addition to the original training image, perhaps from later in the summer melt season. |
| 237 - 245 | Naming the clusters seems to be inconsistent throughout the paper. I recommend using a consistent cluster abbreviation (C1) and name (dry snow/ice) throughout the paper and all the figures to make it easier to follow. Some surface type classifications are introduced in this paragraph without a clear indication of where you got the |
| 241 | Where did the "wet/ridged ice" name come from? Line 186 only said it was similar to "ridged ice", with nothing about wet conditions. |
| 242 | Likewise, the "snow-free/ponded ice" cluster name is not adequately supported by the comparison to the measured spectra. |
| 250 | Use the same names for the clusters in the corresponding figures/text: for example C4 is referred to as snow-free/ponded ice in the text and clean ice in the figure. |
| 272 | "shown here to successfully partition surface reflectance data" implies that there is a measure of success - without clear validation, that seems like the wrong word. |
| 293-294 | While there may not be available in situ data, visible imagery (especially when we get lucky with the clouds!) provides a lot of context with which classification can be achieved: unsupervised ML methods are not the only feasible approach. |
| 307-315 | It is unclear what you mean by "requisite for inferring actual environmental relationships" and the entire next paragraph. Features of the spectral signatures of different surface types seem to be conflated with physical features. Rephrasing might help. |
| 324-325 | Without in situ measurements of freshet water reflectance spectra, how confident are you that this cluster's spectral signature |
| 328 | This approach is limited by the imagery repeat frequency/availability of cloud-free images. |
| 344 | Drainage through strudel holes seems like only one of several possible ways for ponded water to drain in a deformed coastal ice environment. |
| 349 | How much of the names for the different clusters comes from the seasonal progression versus the spectral signatures? |
| 354 | Much of the identified freshet surface on 5/31 seems like it is thin cloud rather than an actual surface process. |
| 356 | This assumes that the C5 cluster only includes river flooding and not additional surface types. It seems more likely that rotten ice is getting classified as C5 rather than overflow lingering on melting, isolated floes late into the summer. |
| 365 | A lot of interpretation is hanging on the assumption that the cluster centroid spectra and in situ surface reflectance data are conclusively representing the same surface(s) – a discussion of validation for the cluster type assignments would help. |

| | |
|---|---|
| 368-370 | This seems to refer to a mis-classified ice type, but that isn't acknowledged elsewhere in the paper. |
| 371-372 | This caution seems to point to exactly what the authors of this paper did. |
| 385 | Adding the true color image next to the classified training image would help, as would masking out known non-ice features including land. |
| 387 | This indicates that the clustering algorithms are insufficient to the task? |
| 407 | This line somewhat overstates the outcome of this study, given the lack of validation of the classifications. |
| 414 | Please provide a reference. |
| 453 | Strudel holes are not necessarily the only drainage mechanism. |
| 481 | This assumes that there is sufficient time-resolution, cloud-free skies, and that the algorithm can successfully and consistently classify surface types. |
| 498 | The proposed operational uses require a level of reliability and certainty that the approach shown in this paper does not demonstrate. While the discussion includes a lot of potential avenues, it is highly speculative. |